# Tetrahydrocannabivarin (THCV) Protects Adipose-Derived Mesenchymal Stem Cells (ASC) against Endoplasmic Reticulum Stress Development and Reduces Inflammation during Adipogenesis

**DOI:** 10.3390/ijms24087120

**Published:** 2023-04-12

**Authors:** Anna Kowalczuk, Krzysztof Marycz, Justyna Kornicka, Sylwia Groborz, Justyna Meissner, Malwina Mularczyk

**Affiliations:** 1National Medicines Institute, Chełmska 30/34, 00-725 Warsaw, Poland; 2International Institute of Translational Medicine, Jesionowa 11, 55-114 Malin, Poland; 3Department of Experimental Biology, Faculty of Biology and Animal Science, Wrocław University of Environmental and Life Sciences, Norwida 27B, 50-375 Wrocław, Poland; 4Faculty of Electronics, Photonics and Microsystems, Wrocław University of Science and Technology, Smoluchowskiego 25, 50-372 Wrocław, Poland

**Keywords:** endoplasmic reticulum (ER), Δ9-tetrahydrocannabivarin, adipocyte

## Abstract

The endoplasmic reticulum (ER) fulfills essential duties in cell physiology, and impairment of this organelle’s functions is associated with a wide number of metabolic diseases. When ER stress is generated in the adipose tissue, it is observed that the metabolism and energy homeostasis of the adipocytes are altered, leading to obesity-associated metabolic disorders such as type 2 diabetes (T2D). In the present work, we aimed to evaluate the protective effects of Δ9-tetrahydrocannabivarin (THCV, a cannabinoid compound isolated from *Cannabis sativa* L.) against ER stress in adipose-derived mesenchymal stem cells. Our results show that pre-treatment with THCV prevents the subcellular alteration of cell components such as nuclei, F-actin, or mitochondria distribution, and restores cell migration, cell proliferation and colony-forming capacity upon ER stress. In addition, THCV partially reverts the effects that ER stress induces regarding the activation of apoptosis and the altered anti- and pro-inflammatory cytokine profile. This indicates the protective characteristics of this cannabinoid compound in the adipose tissue. Most importantly, our data demonstrate that THCV decreases the expression of genes involved in the unfolded protein response (UPR) pathway, which were upregulated upon induction of ER stress. Altogether, our study shows that the cannabinoid THCV is a promising compound that counters the harmful effects triggered by ER stress in the adipose tissue. This work paves the way for the development of new therapeutic means based on THCV and its regenerative properties to create a favorable environment for the development of healthy mature adipocyte tissue and to reduce the incidence and clinical outcome of metabolic diseases such as diabetes.

## 1. Introduction

The endoplasmic reticulum (ER) is an intracellular transport system that participates in a number of essential cellular processes, such as protein synthesis, folding and maturation, as well as in many other signaling pathways. The function of this complex system may be partially impaired by genetic or environmental factors, leading to ER stress [1]. Substantial ER stress activates a complex signaling pathway known as the unfolded protein response (UPR), which drives the cell into the apoptotic or autophagic pathways. This process is transduced and executed by three transmembrane proteins: PKR-like ER kinase (PERK), inositol-requiring enzyme 1 (IRE1), and activating transcription factor 6 (ATF6) [2]. Excessive ER stress is directly associated with several diseases, such as diabetes, insulin resistance, inflammation, atherosclerosis, heart diseases, and neurodegenerative disorders such as Alzheimer’s disease [3].

The ER stress dysfunction of adipose tissue (AT) plays a critical role in the development of obesity-associated metabolic disorders such as type 2 diabetes (T2D) through regulation of the metabolism and energy homeostasis [4]. Obesity is a global health problem, and it is expected that by 2025, more than 300 million people will develop T2D as a result of excessive fat accumulation in adipose tissues [5]. In individuals with obesity, AT is characterized by an abnormal production of cytokines and other pro-inflammatory molecules [6]. For adipose tissue to reach its full maturity and develop its endocrine and immunological function, the process of adipogenesis is activated, where precursors defined as adipose-derived mesenchymal stem cells (ASCs) are differentiated into mature adipocytes [7]. Recent studies show that both in mice and humans, there is a correlation between obesity and impaired ASCs migration, proliferation, and angiogenic potential, hence reducing the therapeutic potential of these cells [8]. Therefore, given the association of ASCs with metabolic disorders such as obesity and obesity-related metabolic diseases, methods for positive ASC stimulation and promotion of positive differentiation are being carefully studied [9,10]. Substantial research efforts have also been undertaken to unravel the correlation between increased ER stress in ASC and conditions such as obesity or insulin resistance [11].

The endocannabinoid system (ECS) is a compound cell-signaling system identified in the 1990s during research on the main compound derived from *Cannabis sativa* L.—Δ 9-tetrahydrocannabinol (THC) [12]. ECS consists of the G-protein-coupled cannabinoid receptors type 1 (CB1R) and type 2 (CB2R), their endogenous ligands (endocannabinoids), and the enzymes responsible for their synthesis and degradation. The main endocannabinoids are anandamide (arachidonoylethanolamide, AEA) and 2-arachidonoylglycerol (2-AG) [13]. The endocannabinoid signaling system is located in the body both centrally (brain) and peripherally (adipose tissue, liver, pancreas, skeletal muscle, gastrointestinal track) and is involved in multiple physiological processes, including maintenance of the energy balance by lipid and glucose metabolism [14]. It has been suggested that ECS plays an important role in adipogenesis and lipogenesis, participating in the development of obesity and type 2 diabetes [15]. It has been shown that CB1 receptor stimulation and its increased mRNA levels lead to inflammation and free fatty acid (FFA) accumulation in cultured adipocytes [14]. Since ECS receptors are expressed in ASCs, their proliferation, differentiation, and overall plasticity can be modulated by either the activation or inhibition of CB1 or CB2, thus modulating the activity and morphology of the mature adipose tissue.

*Cannabis sativa* is a distinctive plant mostly known for its primary psychoactive constituent delta-9-tetrahydrocannabinol (THC). Above that, the plant comprises about 100 different phytocannabinoids including Δ9-tetrahydrocannabivarin (Δ9-THCV), cannabinol (CBN), cannabidiol (CBD), cannabidivarin (CBDV), cannabigerol (CBG), and cannabichromene (CBC). Due to their ability to interact with CB1 and CB2 receptors as their antagonists and inverse agonists, cannabinoids show broad prospects as therapeutic factors in many metabolic diseases, such as obesity, insulin resistance, multiple sclerosis, anorexia, inflammation, epilepsy, schizophrenia, glaucoma, and Parkinson’s and Alzheimer’s disease [16,17]. Among the cannabinoid compounds, the non-psychoactive Δ9-tetrahydrocannabivarin (Δ9-THCV) has gained special interest for its unique properties and non-psychoactive effects, which set it apart from its psychoactive analog, THC. Recent studies indicated that THCV-CB1 receptor antagonists/inverse agonists ameliorated insulin sensitivity and improved glucose tolerance in two mouse models of obesity [18], reduced accumulated lipid levels in vitro in a hepatosteatosis model and in adipocytes [19], and decreased fasting plasma glucose with improved pancreatic β-cell function in a randomized, double-blind, placebo-controlled study on type 2 diabetes [20]. Nonetheless, the anti-obesity feature of THCV is still not well understood, and further research is necessary to confirm the potential therapeutic properties of Δ9-THCV for the treatment of obesity, metabolic syndrome and type 2 diabetes.

In the present work, we introduce an alternative approach based on a pretreatment with Δ9-tetrahydrocannabivarin to overcome issues related to the molecular impairment of ASC and subsequent functions of mature adipocytes. The aim of this strategy is to prevent ER stress and inflammation and its downstream harmful effects, including the development of obesity-associated metabolic disorders such as type 2 diabetes (T2D).

## 2. Results

### 2.1. Evaluation of Morphology and Proliferation Rate

Cell morphology in control and experimental (TUN and THCV) conditions was assessed by confocal microscopy. In this work, we used tunicamycin to induce ER stress; tunicamycin is an inhibitor of UDP-N-acetylglucosamine-dolichol phosphate N-acetylglucosamine-1-phosphate transferase (GPT), therefore blocking the initial step of glycoprotein biosynthesis in the ER. This leads to an accumulation of unfolded glycoproteins in the ER, leading to ER stress. We observed that cells treated with tunicamycin presented deformed cell nuclei with an irregular size and with a more condensed chromatin (Figure 1A). The distribution of F-actin, assessed with the phalloidin staining, was also altered in the tunicamycin-treated cells (Figure 1A). We found reduced actin fibers with the abnormal organization in the TUN group in comparison to the control group. Moreover, the number of mitochondria were reduced and localized near the nuclei compared to the control group, where the mitochondrial network was expanded (Figure 1A). Interestingly, pre-treatment with different doses of THCV successfully protected cells against ER stress, as evidenced by the re-establishment of the mitochondria network, the cytoskeleton integrity and the morphology of the cell nuclei (Figure 1A). We also observed that different concentrations of THCV in the range of 0–10 μM did not affect either cell viability or population doubling time (Figure 1B,C); however, THCV used at 50 and 100 μM reduced cell viability by up to 40% (Figure 1B).

To further study the effect of THCV on cell proliferation, we performed the scratch assay, which showed that the pre-treatment of HuASCs with THCV at concentrations of 1 µM and 5 µM promoted cell migration after 12 h compared to the TUN group (Figure 1D); after 48 h of treatment, cell expansion in the THCV1 and THCV5 groups was around 90% compared to cells treated with tunicamycin (5%) (Figure 1D). Intriguingly, the β-galactosidase activation assay in the HuASCs revealed that THCV reduced the number of age-senescence cells compared to the control group and the tunicamycin-treated group (Figure 1E). Additionally, we investigated the influence of different doses of THCV on the cell proliferation of the HuASCs by quantifying the proliferation marker KI-67. The immunostaining experiment revealed a reduction of the number of KI-67-positive cells in the TUN group compared to the non-treated group (Figure 1F). Interestingly, pre-treated cells with 1 μM THCV presented similar levels of KI-67-positive cells compared to cells treated with tunicamycin (Figure 1F); however, a higher dose of THCV (5 μM) appears in the image more visibly, which may indicate a greater amount of KI-67-positive cells compared to the TUN-treated and non-tread group (Figure 1F).

We then analyzed the effect of THCV in the cell’s colony-forming capacity as experimental evidence of the proliferation-inducing effect of THCV. We observed that tunicamycin inhibited cell proliferation and the ability to form colonies, while pre-treatment with THCV in any tested concentration enhanced cell division, evidenced by the higher number of CFUs detected in the clonogenic assay (Figure 1G).

In order to study cell proliferation at the transcriptional level, the relative expression of miRNA involved in cell proliferation was assessed by qRT-PCR. Results showed that in the tunicamycin-induced ER stress group, the expression of miR101 1/2 was significantly enhanced in comparison to the experimental and control groups (Figure 1H). The opposite phenomenon was noticed in the expression of miR17, where transcript levels were upregulated not only in the TUN group, but also in the THCV groups (Figure 1I).

The data obtained indicate that THCV used in concentrations of 1 μM and 5 μM promotes cell proliferation, as well as their migration and clonogenic potential.

### 2.2. Evaluation of Apoptosis

To establish whether pre-treatment with THCV modulates cell death, the gene expression of some apoptotic markers was assessed by qPCR. Results clearly showed that THCV reduced *p53* expression compared to the TUN group (Figure 2A). Moreover, we observed that *p21* was upregulated upon treatment with THCV 1 μM, while THCV at 5 μM reduced its expression levels, with no differences compared to the TUN condition (Figure 2B). Regarding the gene expression of *Casp9*, it was increased in response to the TUN group, but the addition of THCV 5 μM reestablished the control levels (Figure 2C). Furthermore, the results showed an increased expression of *BCL2* in cells pre-treated with different doses of THCV (Figure 2D). Interestingly, we observed an upregulation of *BAX* in both the TUN and THCV 1 μM groups compared to THCV 5 μM (Figure 2E). In order to confirm the role of THCV in the modulation of the apoptotic cell death pathway, we performed an immunostaining of *Casp3*, observing an increased number of Casp3-positive cells in the TUN group compared to the control group (Figure 2G). Most importantly, THCV at 1 μM and 5 μM decreased the *Casp3* signal, indicating that THCV partially protects from tunicamycin-induced apoptosis (Figure 2G). This finding was supported by the data obtained from the qRT-PCR analysis and the Western blot analysis, where the expression of caspase-3 was decreased by THCV at the gene (Figure 2H) and protein level (Figure 2I). To support the results obtained in the apoptosis experiments, we performed the Muse^®^ Annexin V & Dead Cell analysis and the Muse^®^ MultiCaspase analysis, which showed significant differences in the percentage of live cells between TUN and the THCV conditions (Figure 2J,K). Additionally, we found that 10% of the cells were caspase-3-positive in the tunicamycin-treated group, while only 1% were caspase-positive cells in the THCV group (Figure 2J,K). THCV treatment of HuASCs reduced the caspase-3-positive cells from 15% to 5% (## *p* < 0.01 for THCV1 and ### *p* < 0.001 for THCV5).

### 2.3. Evaluation of ER Stress

We then aimed to evaluate the effect of THCV on tunicamycin-induced ER stress with different approaches. First, we found that tunicamycin induced a rearrangement of the ER network and the nuclei shape compared to the control condition, while pre-treatment with THCV partially restored these tunicamycin-induced effects (Figure 3A). These results were supported by the observation of the upregulation of the UPR-genes *PERK*, *IRE*, *ATF6*, *eIF2-α*, *CHOP* and *XBP1* by qRT-PCR (Figure 3B–G). It is important to note that *PERK* expression was decreased in the THCV pre-treated condition compared to the TUN group (Figure 3B). Interestingly, the expression of *IRE* and *ATF6* was downregulated to a greater extent in the THCV 1 μM condition compared to the THCV 5 μM group (Figure 3C,D). Furthermore, we found that THCV application significantly reduced *CHOP*, *XBP* and *eIF2-α* gene expression compared to the TUN condition (Figure 3E–G). These findings were in accordance with the protein expression analysis, where we found that the *eIF2-α* protein was upregulated in the TUN group compared to the control and subsequently downregulated in the presence of THCV (Figure 3H).

### 2.4. Evaluation of Inflammation

We then sought to study the inflammatory response in native adipogenesis conditions by analyzing the gene expression of different biomarkers. We found that the expression of the anti-inflammatory genes *TGFβ* and *IL-10* was decreased in response to TUN and THCV (Figure 4A,B). Interestingly, *IL-13* expression was upregulated in the TUN group, and treatment with THCV reestablished the control condition expression level (Figure 4C). Importantly, we found that tunicamycin increased *IL-1β* and *IL-6* gene expression, and treatment with THCV at 5 μM decreased the tunicamycin-induced *IL-1β* (Figure 4D,E). Surprisingly, *IL-6* gene expression in THCV 5 μM was downregulated compared to the TUN condition, which increased the basal *IL-6* expression levels (Figure 4F). However, the analysis of the expression at the protein level revealed that *IL-6* was downregulated upon pre-treatment with THCV at 5 μM (Figure 4G).

Additionally, micro-RNA expression was evaluated. Intriguingly, in the THCV pre-treated group, the relative expression of miR 16-5p was increased compared to the TUN group (Figure 4H). Furthermore, in each group, we observed an increased expression of miR203b (Figure 4I). Interestingly, in the TUN group, miR21 expression was downregulated, but in the THCV 1 μM and THCV 5 μM group, it was increased (Figure 4J). We also observed an upregulated expression of miR24-3p for both concentrations (Figure 4K). The same phenomenon as that in miR21was observed in miR146-5p expression (Figure 4L).

Most importantly, the expression of the pro-inflammatory genes TNF-α (Figure 4M) and IL-6 (Figure 4N) as well as the protein levels of IL-6 (Figure 4O) were decreased after pre-treatment with THCV. IL-4 was increased in the adipogenic conditions when treated with tunicamycin, while pre-treatment with THCV 5 μM decreased the expression of that gene (Figure 4P).

## 3. Discussion

The pharmacological and prophylactic application of cannabinoids has a huge therapeutic potential in degenerative diseases of the nervous system and blood system and in obesity. However, the effects of individual compounds isolated from marijuana are not well understood [18,21,22]. In this study, the protective properties of THCV were investigated. Here, we describe very promising effects of THCV on HuASCs morphology, expansion, cell senescence, apoptosis, inflammation and protection against ER stress. Moreover, we found that the pre-treatment of THCV has strong anti-inflammatory properties in cells during adipogenesis, which may prove to be a crucial and fundamental aspect in the treatment of obesity.

It is been widely known that ER stress induced with tunicamycin stimulates the cellular response (UPR), leading to cell dysfunction, including proliferation, inflammation, and apoptosis. We found that the pre-treatment of HuASCs with THCV protects against the harmful effects of ER stress while preserving natural cell morphology, similar population doubling time (PDT) and expansion properties. In fact, we revealed that THCV protects cells against senescence and stimulates cells for increased proliferation and colony formation. Interestingly, the opposite phenomenon was shown by Hohmann T et al. [23], where they observed that the application of cannabinoids had an influence on cell motility, morphology and actin organization in cancer cells, showing anti-cancer properties. Our data indicate that cannabinoids, and in particular THCV, have a huge protective potential in terms of increasing the ability of MSCs to proliferate and migrate, which is necessary during the regeneration of damaged cells in obesity and other degenerative diseases. To our knowledge, no study so far has been presented on THCV’s effects on ER-stress-induced MSC. However, we speculate that the obtained data in this study are the result of cell type and are closely related to the presence of CB1 receptors on the surface of specialized cells [24].

One of the most important consequences of the disruption of cellular homeostasis caused by tunicamycin is increased cellular stress and the activation of the apoptosis pathway. As we have shown, the implementation of higher doses of the cannabinoid prior to the application of tunicamycin resulted in reduced levels of pro-apoptotic mRNAs, in particular *p53*, *p21*, *BAX* and caspases (*Casp3*, *Casp6* and *Casp9*), known apoptosis activators. Moreover, we showed a reduced percentage of caspase-3-positive cells as well as a reduced percentage of early and late apoptotic HuASCs at both 1 µM and 5 µM, further supporting the protective properties of THCV. On the other hand, according to the latest data, cannabinoids have strong anti-cancer properties, which are revealed by the induction of pro-apoptotic genes in many cancer lines [23,25,26]. These reports make THCV a desirable compound in the fight against altered cancer cells, but as our research shows, THCV can promote cell survival and ER stress protection for stem cells at the same time, making it a compound with high therapeutic potential and a broad spectrum of activity.

Recent studies have shown that endoplasmic reticulum stress and activation of the unfolded protein response (UPR) promotes obesity and insulin signaling disturbance [27]. Indeed, ER stress and UPR activation can cause damage at the beginning of adipose tissue formation, i.e., in the HuASCs, by activating proinflammatory cytokines [28]. Thus, altered progenitor function may disturb their regenerative potential. On the other hand, this can be can be a novel way for targeting those cells as therapy targets in metabolic disorders [29].

Here, we investigated the impact of human adipose-delivered stem cells pre-treated with THCV to evaluate their influence on ER stress and inflammation-related genes. Moreover, we found that the expression levels of the key UPR-related factors *PERK, IRE, ATF6*, *CHOP, XBP1* and *eIF2-α* were reduced in the groups pre-treated with the cannabinoid compound. This finding is in agreement with our previous studies with another cannabinoid—CBD (cannabidiol) [30]. The results obtained when THCV was used at 1 µM and 5 µM were similar between genes, suggesting that both of the concentrations are similarly effective. Although the correlation between ER stress signaling and the cannabinoid system is still not well understood, our research shows for the first time that THCV can protect the ER in HuASCs from degenerative pathway activation and can regulate key ER stress factors. Moreover, we evaluated sufficient markers of the chronic inflammation state, which usually occurs simultaneously with ER stress and activation of the unfolded protein response in many diseases, including diabetes and obesity [31]. The inflammatory status of HuASCs can contribute to pathological changes in adipocytes, resulting in the secretion of adipokines and proinflammatory cytokines, thus disturbing their beneficial regenerative and immunomodulatory properties [32]. Many studies have been published related to this, including our own that focus on improving the potential of HuASCs in regenerative medicine, including low-energy extracorporeal shock wave therapy [33], external magnetic field [34], 5-Azacytydine and resveratrol treatment [35], astaxanthin treatment [36] and many more. Herein, we proposed THCV as another factor for the retention of the unfavorable microenvironment of inflamed HuASCs. While the effect on the inflammatory cytokines in native HuASCs were not significant, we observed that THCV in a 5 µM concentration can effectively retain the expression of pro-inflammatory cytokines during adipogenesis. These findings correspond with other research that revealed that THCV suppresses signs of inflammation and inflammatory pain in mice [37]. On the other hand, we performed PCR analysis for key micro-RNAs involved in inflammation. Pre-treatment with THCV in a 5 µM concentration significantly upregulated the expression of miR 16-5p, which can inhibit the expression of *IL-6* and *TNF-α* as well as apoptosis [38]. Similarly, we observed that the overexpression of miR-203b has a beneficial effect on metabolic homeostasis [39]. We have also found an increase in the expression of miR-21, miR-24-3p and miR-146-5p, which are involved in the activation of a pro-inflammatory macrophage phenotype and the attenuation of inflammation [40,41,42].

These results strongly suggest that THCV can effectively reduce inflammation in tunicamycin-impaired HuASCs; nevertheless, there is still a need for the further examination of detailed Δ9-Tetrahydrocannabivarin mechanisms of action. For a better understanding of THCV’s impact on adipose tissue development, we examined the inflammatory gene expression levels during adipogenesis. We observed significant downregulation of the key pro-inflammatory cytokines *IL-6*, *TNF-α* and *IL-4* for HuASCs pre-treated with THCV 5 µM before adipogenesis induction. Surprisingly, for the same groups, we noticed a reduced expression of *IL-4*, which is well known to suppress adipocyte differentiation [43].

In summary, our studies revealed that pre-treatment with THCV can mitigate the negative effects of metabolically impaired HuASCs due to a promoting effect on viability, proliferation, and multipotency, as well as due to the improvement of normal morphology. Indeed, we report that THCV can reduce ER stress and inflammation in HuASCs, which can improve their regenerative properties and create a stable environment for the development of healthy adipocyte tissue, thus preventing metabolic diseases such as diabetes. Our research reveals the great potential of plant cannabinoid; nonetheless, there still is a need for further experiments that could explore the molecular mechanisms and receptors by which THCV modulates HuASCs metabolism. Furthermore, we are planning to expand our research with a detailed adipogenesis panel for a greater understanding of THCV’s impact on the differentiation of adipocyte precursor cells in mature adipose tissue. Considering that nowadays there is still a need for metabolic disorder (including obesity) prevention and the enhancement of regenerative outcomes of autologous stem cells, the potential use of the natural plant compound THCV, which is non-psychotropic, could be an effective and economical way to cope with those obstacles.

## 4. Materials and Methods

### 4.1. Experimental Model Setting

Human adipose-derived stem cells (HuASCs) were seeded onto 24-well plates at a density of 25 × 10^4^/per well. HuASCs were cultured in Dulbecco’s Modified Eagle’s Medium (DMEM, Gibco Carlsbad, CA, USA) containing 1 g/L glucose, supplemented with 10% fetal bovine serum (FBS, Gibco Carlsbad, CA, USA) and 1% penicillin-streptomycin antibiotic solution (Gibco Carlsbad, CA, USA). Cell cultures were pre-treated with THCV (Cayman Chemical, 18091, Ann Arbor, MI, USA) at a concentration of 1 µM and 5 µM for 24 h (h). Then, the medium was replaced by medium containing 5 mmol/mL tunicamycin (Sigma-Aldrich, T7765, Poznan, Poland).

For adipose differentiation, HuASCs were cultured in commercially available medium StemPro^®^ Adipogenesis Differentiation (A1007001, Gibco, Thermo Fisher Scientific, Warsaw, Poland). Cells were seeded onto 24-well plates at a density of 20 × 10^4^/per well. The culture medium containing Adipocyte Differentiation Basal Medium, Adipogenesis Supplement and Gentamicin 5 μg/mL was changed every 3 days. Adipose differentiation was conducted for 14 days; then, cells were used for further experiments.

### 4.2. Proliferation Rate and Scratch Assay

Cell viability was evaluated with a resazurin-based assay (TOX8 In Vitro Toxicology Assay Kit, Sigma Aldrich, Poznan, Poland). After 24 h of incubation with THCV at a concentration range of 0–100 μM, culture medium was replaced by medium containing DMEM, 10% FBS, 1% PS and 10% *v*/*v* resazurin dye. Cells were incubated for 4 h at 37 °C. The post-cultured medium was transferred to a 96-well plate in a volume of 100 µL in triplicate. The fluorescence was measured at 600 nm and 690 nm as a reference wave. Population Doubling Time (PDT) was calculated using an online algorithm (http://www.doubling-time.com/compute.php (accessed on 10 October 2022)).

In order to test the ability of cells to form colonies, a clonogenic assay was performed. For this purpose, cells were seeded onto a 6-well plate at a density of 1 × 10^2^/per well. Cells were treated with THCV and/or tunicamycin. After 7 days of incubation, cells were fixed with cold 4% PFA (Sigma-Aldrich, P6148), and then colonies were stained with pararosaniline (Sigma-Aldrich, P3750). A series of photos were taken via phone. Colony-forming unit fibroblastic assays (CFU-Fs) were analyzed using the formula described by Kornicka et al. [35].

A scratch assay was performed to evaluate the ability of the cells to migrate. Cells were seeded onto a 96-well plate at a density of 10 × 10^4^/per well. HuASCs were treated with compounds as described above. Then, a horizontal line (scar) in the center of the 96-well plate was made using a 10 μL pipette tip. The pictures were taken with the Zoe Fluorescent Cell Imager after 0 h, 12 h, 24 h and 48 h after scarring. Data were analyzed with ImageJ (version 1.53t, Bethesda, MD, USA) and GraphPad Prism 8 Software (San Diego, CA, USA).

### 4.3. Visualization of Cell Organelles

A confocal microscope (Observer Z1 Confocal Spinning Disc V.2 Zeiss) was used for the visualization of the cell nucleus, mitochondria, cytoskeleton and the endoplasmic reticulum. The endoplasmic reticulum was stained with 1 µM ER-Tracker™ Green (Invitrogen™, E34251, Thermo Fisher Scientific, Warsaw, Poland). For this, cell medium was changed to Hank’s Balanced Salt Solution with calcium and magnesium containing 1 µM ER-Tracker™ Green. After 30 min (min) of incubation at 37 °C, cells were fixed with cold 4% paraformaldehyde for 30 min. Mitochondria were stained with 100 nM MitoRed dye (Sigma-Aldrich, 53271, Poznan, Poland) on viable cells for 30 min in the dark at 37 °C. Medium containing the MitoRed dye was removed, and cells were washed three times with PBS. Cells were fixed with 4% PFA as described above and permeabilized with 0.1% Triton X-100 (Sigma-Aldrich, 93443, Poznan, Poland) solution for 20 min. The cytoskeleton was stained using atto-488-labeled PI (Sigma-Aldrich, 49409, Poznan, Poland) (1:800 in PBS) for 45 min in the dark at RT. Cell nuclei were stained with DAPI (Invitrogen™, Warsaw, Poland), following the instructions of the manufacturer (Faramount Aq Mounting Medium, Dako).

### 4.4. Immunostaining with KI-67

To visualize the proliferation marker Ki-67, immunostaining with Ki-67 antibody (Abcam, ab15580, Cambridge, UK) was performed. HuASCs were treated with the appropriate compounds for 24 h; then, the medium was removed and cells were fixed with 4% PFA as described above. Samples were washed with PBS twice and permeabilized with 0.05% Triton X-100 in PBS for 15 min at room temperature in the dark. Treated and non-treated cells were incubated overnight with the Ki-67 antibody diluted in 10% Goat Normal Serum (Invitrogen, #31872, Warsaw, Poland) in PBS at 4 °C (1:1000). Cells were washed with PBS three times and incubated with atto-594 secondary antibody (Sigma-Aldrich, 77671) diluted in PBS for 1 h in the dark at RT (1:1000). Cell nuclei were stained with DAPI (Faramount Aq Mounting Medium, Dako). Cells were visualized with a confocal microscope (Observer Z1 Confocal Spinning Disc V.2 Zeiss, Germany), and the obtained data were analyzed using Image J Software (Bethesda, MD, USA).

### 4.5. Evaluation of β-Galactosidase Activation

Cell senescence is a process characterized by a decreased rate of cell division and structural changes in the cell morphology. In order to investigate the senile cell processes, a test with Senescence Cells Histochemical Staining Kit (Sigma Aldrich, Poznan, Poland) was performed according to the manufacturer’s instructions. Cells were treated with 1× Fixation Buffer for 6 min, incubated in Staining Mixture at 37 °C overnight, and visualized with an invert microscope (Leica DM1000 LED, Wetzlar, Germany).

### 4.6. Gene Expression Analysis

To evaluate the gene expression of the selected biomarkers (Table 1), cells were homogenized using EXTRazol (Blirt, Gdańsk, Poland) according to the instructions provided by the manufacturer. Total RNA was diluted in DEPC-water, and its concentration and purity were measured using a nanospectrophotometer (Epoch, BioTek, Janki, Poland). A total of 150 ng of RNA were used to synthesize cDNA by using the Takara PrimeScriptTM RT Reagent Kit with gDNA Eraser (Perfect Real Time)(Biokom, Janki, Poland). Real-Time PCR was performed using the SensiFast SYBR & Fluorescein Kit (Bioline, London, UK) according to the instructions provided by the manufacturer. The Real-Time PCR program was set as follows: 95 °C for 2 min, followed by 41 cycles at 95 °C for 15 s, annealing for 30 s and elongation at 72 °C for 15 s. The qPCR results were replicated in 3 independent experiments, and data were statistically analyzed. Relative gene expression was normalized by the housekeeping gene glyceraldehyde 3-phosphate dehydrogenase (GAPDH) using the 2^−ΔΔCT^ method.

In order to analyze miRNA expression, the Mir-X miRNA First Strand Synthesis Kit (Takara, Biokom, Janki, Poland) was used. Briefly, gDNA traces were removed by incubating the RNA with the DNase I at 37 °C for 30 min. Then, RNA was mixed with mRQBuffer (2X) and mRQEnzyme. The reaction mixture was incubated at 37 °C for 1 h, then at 85 °C for 5 min. The expression level of miRNA was analyzed by Real-Time PCR using the MicroRNA first-strand synthesis kit according to the instructions provided by the manufacturer. Briefly, the reaction mixture contained water, SensiFast SYBR & Fluorescein Kit (Bioline, London, UK), miRNA-specific primer (Table 2), mRQ 3’primer, and cDNA. As a reference sample, U6F primer and U6R primer were used.

### 4.7. Proteins Profiling Using Western Blot

In order to evaluate protein levels in cells, Western blot analysis was conducted. After 24 h of treatment under different conditions, HuASCs were harvested and homogenized in RIPA buffer on ice. Protein concentrations were verified using the Pierce™ BCA Protein Assay Kit (Life Technologies, Carlsbad, CA, USA). A final concentration of 25 µg of protein was used for each sample and denatured in a 4 × Laemmli loading buffer (Bio-Rad, Hercules, CA, USA) for 5 min at 95 °C prior to electrophoresis. Denatured proteins were separated by SDS-PAGE electrophoresis at 100 V for 90 min in Tris/glycine/SDS buffer and transferred onto polyvinylidene difluoride (PVDF) membranes (Bio-Rad, USA) at 100 V, 250 mA for 1 h at 4 °C in a Tris/glycine buffer/methanol as described previously in Malicka A. et al. [44]. The obtained membranes were then blocked in a 5% non-fat milk solution in TBST for 1 h at room temperature. Membranes were then incubated with the corresponding primary antibody (Table 3) overnight at 4 °C. Excess antibodies were washed with TBST, and membranes were additionally incubated for 1 h at room temperature with HRP-conjugated secondary antibodies (dilution 1:1000 in TBST, Sigma). The chemiluminescent signals were monitored with the ChemiDoc MP Imaging System (Bio-Rad, USA), and the results were analyzed with Image Lab Software (Bio-Rad, Hercules, CA, USA).

### 4.8. Cell Cycle Analysis

The analysis of the cell cycle was performed by the Muse^®^ Cell Cycle Kit, following manufacturer’s instructions. Cells were suspended in ice cold 70% ethanol and incubated overnight at −20 °C in the dark. Cells were diluted in the Muse^®^ Cell Cycle Reagent and incubated for 30 min in the dark at RT. The cell cycle was tested using Muse™ Cell Analyzer (Merck, Darmstadt, Germany).

### 4.9. Evaluation of Apoptosis

To evaluate cells apoptosis, the Muse^®^ Annexin V & Dead Cell Kit was used, following the manufacturer’s instruction. Treated and non-treated cells were incubated with Muse^®^ Annexin V & Dead Cell reagent in the dark for 20 min at RT. Data were acquired with a Muse™ Cell Analyzer (Merck, Germany).

### 4.10. Statistical Analysis

Obtained data were analyzed by one-way variance analysis (ANOVA) using GraphPad Software 8 (San Diego, CA, USA) according to Tukey test. Statistically significant results (comparison of non-treated cells [CTRL] to cells treated with THCV at a concentration of 1 and 5 μM or/and tunicamycin [TUN]) were indicated with asterisks depicted as follows: *p* < 0.05 (*), *p* < 0.01 (**) and *p* < 0.001 (***). Statistically significant results (comparison of treated cells with THCV [THCV1, THCV5] to cells treated with tunicamycin) were indicated with a hash, depicted as follows: *p* < 0.05 (#), *p* < 0.01 (##) and *p* < 0.001 (###). Data were obtained from at least three independent experiments and represented as mean ± standard deviation (SD).

## Figures and Tables

**Figure 1 ijms-24-07120-f001:**
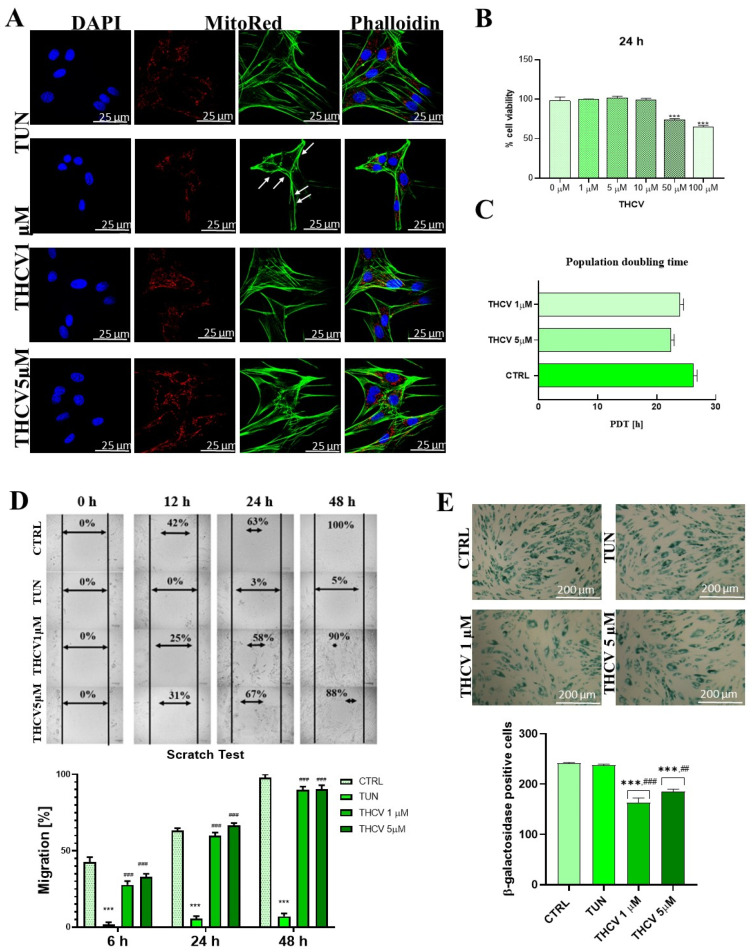
Effects of pre-treatment of THCV on HuASCs. (**A**) Morphological changes of the cells were visualized by confocal microscopy. Cells were stained with DAPI (nuclei), phalloidin (F-actin), and MitoRed dye (mitochondria) (**B**). The changes in cytoskeleton caused by tunicamycin are marked with white arrow. Cell viability was analyzed by the resazurin-based assay. (**C**) Population doubling time was determined in the shown conditions. (**D**) The scratch assay and (**G**) clonogenic assay were performed. (**E**) Cell senescence was estimated by the β-galactosidase activation assay. (**F**) The expression of the proliferation marker KI-67 was assessed with a specific antibody and visualized by confocal microscopy. (**H**,**I**) Expression of miR 101 1/2 and miR17 was investigated with qRT-PCR. Data are expressed as mean ± SD. Statistical significance is indicated with an asterisk (*) when comparing the results to the CTRL condition and with a hashtag (#) when compared to the TUN condition; they are depicted as follows: * *p* < 0.05, ## *p* < 0.01, ***, ### *p* < 0.001.

**Figure 2 ijms-24-07120-f002:**
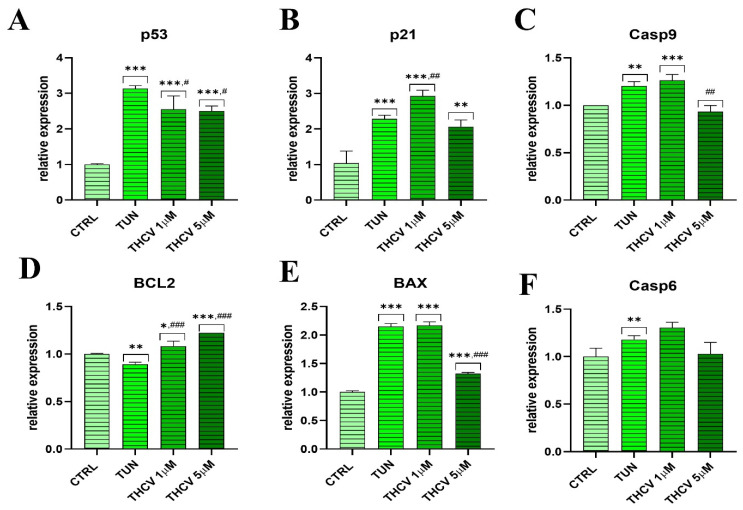
THCV protect HuASCs against apoptosis. (**A**–**F**,**H**) Relative expression of apoptosis-related genes (*p53*, *p21*, *Casp9*, *BCL2*, *BAX*, *Casp6*, *Casp3*) was determined by qRT-PCR. (**G**) Immunostaining of caspase-3 with a specific antibody. (**I**) Caspase 3 (*Casp3*) protein expression by Western blot. (**J**) Muse^®^ Annexin V & Dead Cell analysis and (**K**) Muse^®^ Multicaspase to quantify cell viability and apoptotic cells. Data are expressed as mean ± SD. Statistical significance indicated with an asterisk (*) when comparing the results to the CTRL condition and with a hashtag (#) when comparing to the TUN condition; they are depicted as follows: *, # *p* < 0.05, **, ## *p* < 0.01, ***, ### *p* < 0.001.

**Figure 3 ijms-24-07120-f003:**
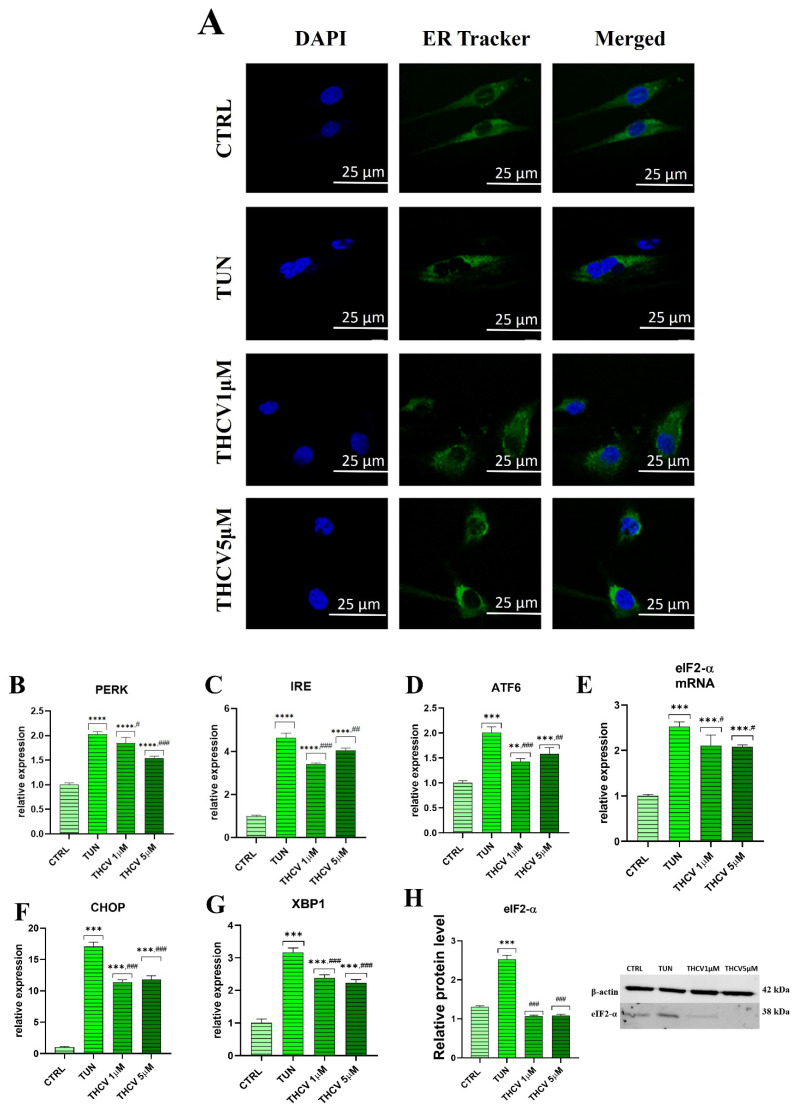
THCV reduce ER stress in HuASCs. (**A**) Representative ER Tracker images of the endoplasmic reticulum staining under the indicated conditions. (**B**–**G**) Relative expression of genes related to ER stress (*PERK*, *IRE*, *ATF-6*, *eIF2-α*, *CHOP* and *XBP1*) was assessed with qRT-PCR. (**H**) The relative protein level of *eIF2-α* was quantified by Western blot. Data are expressed as mean ± SD. Statistical significance indicated with an asterisk (*) when comparing the results to the CTRL condition and with a hashtag (#) when comparing to the TUN condition; they are depicted as follows: # *p* < 0.05, **, ## *p* < 0.01, ***, ### *p* < 0.001, **** *p* < 0.0001.

**Figure 4 ijms-24-07120-f004:**
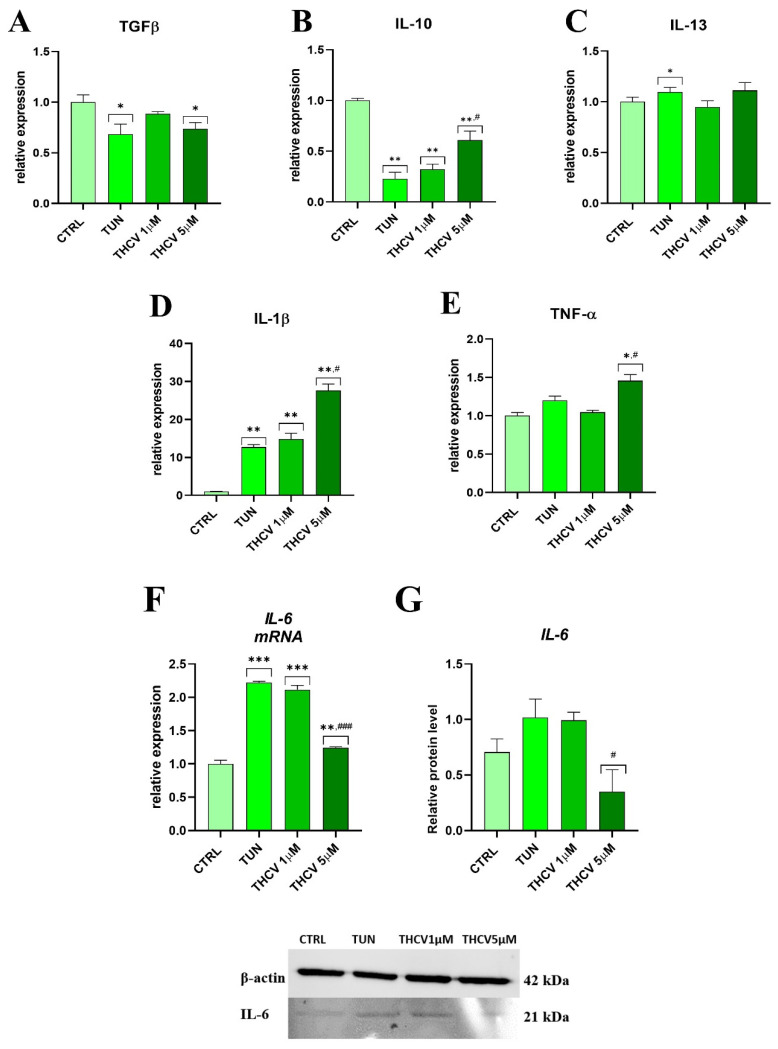
THCV mitigate inflammation in human adipose-derived stem cells. (**A**–**F**) Analysis of the gene expression of pro-inflammatory and anti-inflammatory genes by qRT-PCR (TGFB, IL-10, IL-13, IL-1b, IL-6, TNFa). (**G**,**O**) Quantification of IL-6 protein level by Western blot. (**H**–**L**) Relative expression of microRNA by qRT-PCR (miR16-5p, miR203b, miR21, miR24-3p, miR146-5p). (**M**,**N**,**P**) Analysis of the gene expression of inflammatory factors by qRT-PCR (IL-4, IL-6, and TNFa). Data are expressed as mean ± SD. Statistical significance indicated with an asterisk (*) when comparing the results to the CTRL condition and with a hashtag (#) when comparing to the TUN condition; they are depicted as follows: *, # *p* < 0.05, **, ## *p* < 0.01, ***, ### *p* < 0.001, **** *p* < 0.0001.

**Table 1 ijms-24-07120-t001:** Sequences of primers used in qPCR.

Gene	Primer Sequence (5′->3′)
*BAX*	F: ACCAAGAAGCTGAGCGAGTGTC
	R: ACAAAGATGGTCACGGTCTGC
*BCL2*	F: ATCGCCCTGTGGATGACTGAG
	R: CAGCCAGGAGAAATCAAACAGAGG
*p21*	F: TGCCGAAGTCAGTTCCTTGT
	R: GTTCTGACATGGCGCCTCC
*p53*	F: AGTCACAGCACATGACGGAGG
	R: GGAGTCTTCCAGTGTGATGATGG
*Casp3*	F: GCGGTTGTAGAAGTTAATAAAGGT
	R: CGACATCTGTACCAGACCGAG
*Casp6*	F: TCATGAGAGGTTCTTTTGGCAC
	R: CACACACAAAGCAATCGGCA
*Casp9*	F: TTGGTGATGTCGAGCAGAAAG
	R: CCAGGGTCTCAACGTACCAG
*GPX*	F: CTCCGGAACAACAGCCTTCT
	R: GGAAAGGGGTCTGTGATGGG
*SOD1*	F: GACCATTGCATCATTGGCCG
	R: CAAGCCAAACGACTTCCAGC
*SOD2*	F: GGAGCGGCACTCGTGG
	R: CAGATACCCCAAAGCCGGAG
*SIRT1*	F: ACAGGTTGCGGGAATCCAAA
	R: GTTCATCAGCTGGGCACCTA
*IL-1β*	F: AAACAGATGAAGTGCTCCTTCCAGG
	R: TGGAGAACACCACTTGTTGCTCCA
*IL-6*	F: TCCTTCTCCACAAACATGTAACAA
	R: ATTTGTGGTTGGGTCAGGGG
*TNFα*	F: AGTGACAAGCCTGTAGCCCA
	R: GTCTGGTAGGAGACGGCGAT
*IL-4*	F: CTTTGCTGCCTCCAAGAACAC
	R: GCGAGTGTCCTTCTCATGGT
*IL-10*	F: AGACAGACTTGCAAAAGAAGGC
	R: TCGAAGCATGTTAGGCAGGTT
*PERK*	F: TGCTCCCACCTCAGCGAC
	R: TTTCAGGATCCAAGGCAGCA
*eIF2-α*	F: ATGTTTCAGCCAAGCCCAGA
	R: ACCAGGGGATCTACCACCAA
*CHOP*	F: TAAAGATGAGCGGGTGGCAG
	R: GGATAATGGGGAGTGGCTGG
*ATF6*	F: ACCTCCTTGTCAGCCCCTAA
	R: CACTCCCTGAGTTCCTGCTG
*IRE1*	F: CGGCCTCGGGATTTTTGGA
	R: AGAAAGGCAGGCTCTTCCAC
*XBP1*	F: CGCGGATCCGAATGAAGTGAGGCCAGTG
	R: GGGGCTTGG TATATATGTGG

*BAX*: BCL-2-associated X protein; *BCL2*: B-cell lymphoma 2; *p21*: cyclin-dependent kinase inhibitor 1A; *p53:* tumor suppressor p53; *Casp3*: Caspase-3; *Casp6*: Caspase-6; *Casp9*: Caspase-9; *GPX*: Glutathione peroxidase; *SOD1*: Superoxide dismutase [Cu-Zn]; *SOD2*: Superoxide dismutase 2; *SIRT1*: Sirtuin 1; *IL-1β*: Interleukin 1β; *IL-6*: Interleukin-6; *TNFα*: Tumor necrosis factor α; *IL-4*: Interleukin-4; *IL-10*: Interleukin-10; *PERK*: Protein Kinase RNA-like ER Kinase; *eIF2-α*: Eukaryotic Initiation Factor 2 α; *CHOP*: C/EBP homologous protein; *ATF6*: Activating Transcription Factor 6; *IRE1*: Inositol-requiring enzyme 1; *XBP1*: X-box binding protein 1.

**Table 2 ijms-24-07120-t002:** Sequences of microRNA primers used in qPCR.

Primer miRNAs	Primer Sequence (5′->3′)
miR101-1/2	TACAGTACTGTGATAACTGAA
miR17-5p	CAAAGTGCTTACAGTGCAGGTAG
miR16-5p	TAGCAGCACGTAAATATTGGCG
miR-203b	TTGAACTGTTAAGAACCACTGGA
miR-21	TAGCTTATCAGACTGATGTTGA
miR 24-3p	TGGCTCAGTTCAGCAGGAACAG
miR 146-5p	TGAGAACTGAATTCCATGGGTT

miR: micro RNA.

**Table 3 ijms-24-07120-t003:** List of antibodies used in study.

Antibodies	Concentrations	CAT Numbers	Company
*β-actin*	1:1000	orb10033	Biorbyt (Cambridge, UK)
*Casp3*	1:1000	c8487	Sigma (Poznan, Poland)
*IL-6*	1:1000	ab6672	Abcam (Cambridge, UK )
*eIF2-* *α*	1:500	nbp2-67353	Novus (Centennial, CO, USA)

*Casp3*: Caspase-3; *IL-6*: Interleukin 6; *eIF2-α*: Eukaryotic Initiation Factor 2 α.

## Data Availability

The data that support the findings of this study are available from the corresponding author, upon reasonable request.

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
