# Peer review of "Tetrahydrocannabivarin (THCV) Protects Adipose-Derived Mesenchymal Stem Cells (ASC) against Endoplasmic Reticulum Stress Development and Reduces Inflammation during Adipogenesis"

_ijms, 2023, doi:10.3390/ijms24087120_

Round 1
Reviewer 1 Report
The authors Kowalczuk et. al. in this paper discuss the critical role of the endoplasmic reticulum (ER) in cell physiology and the link between ER stress in adipose tissue and metabolic disorders such as type 2 diabetes. They investigate the protective effects of Δ9-tetrahydrocannabivarin (THCV), a cannabinoid compound isolated from Cannabis sativa L., against ER stress in adipose-derived mesenchymal stem cells. Their results demonstrate that THCV prevents subcellular alterations, restores cell migration and proliferation, and reduces the expression of genes involved in the unfolded protein response pathway, which were upregulated upon induction of ER stress. This study highlights the potential therapeutic use of THCV in reducing the incidence and clinical outcomes of metabolic diseases such as diabetes.
The reviewer feels the manuscript warrants a major revision to be considered for publication. Following concerns need to be addressed.
Minor
1. Fig 1 A. HCV scale bar is missing plus it overlaps with figure below it.
2. Fig 1 D. Representative image: mention the THCV concentrations?
3. Figures aren’t arranged in the order of their referencing in the manuscript. Fig 2 H and I have been mentioned before Figure 2 A – G. Jump from figure 1 G to 2 H? Figure referencing wrongly done. I think the author means 1 H and 1 I.
4. What is the sample size (2 A - G)
5. Section 3.4 Figures need to be arranged in the order of their reference
6. Could make sentences concise to enhance readability
eg: Page 1, lines 24 – 26, 249 – 253, lines 484 - 498
7. The paper is riddled with minor grammatical/formatting errors:
Line 136: the horizontal line ïƒ a horizontal line
Line 158: permeabilized
Line 168: decrease ïƒ decreased
Line 177: dilute ïƒ diluted
Line 209: protein’s ïƒ protein
Line 243: hashtag ïƒ hash; also in multiple other places in the manuscript
Lines 253 – 254
Line 265: on proliferation
Line 273: proliferationmarker ïƒ proliferation marker
Line 278: non-tread? ïƒ non-treated
Line 404: comma between references
Line 155, 160 KI67 ïƒ Ki67
Results section: tenses mixed up
Line 263: The authors probably didn’t mean to use double negative ïƒ affected neither (remove not); population doubling time not doubling population time
Line 43; associated with
Line 45 full stop placement
Line 437: studies has ïƒ have
Line 453: pathways ïƒ pathway
Line 452: are still ïƒ is still
Line 454: sate??
8. Line 105, 114, 128, 135: cell concentration? Volume?
9. Line 122: abbreviation used LG, PS (no full form mentioned previously)
Major
1. Fig 1 F. The immunostaining experiment revealed a reduction of the number of KI-67-positive cells in the TUN group compared to the non-treated group (Fig. 1F). ïƒ Can’t say so based off representative image. Levels look similar. More suitable representative images need to be chosen or conclusions need to be carefully made.
2. Statistical significance representation has been unnecessarily complicated. Differences between groups can be indicated by lines being drawn over groups, with asterisks. This will make usage of hash redundant. Referencing the methods section will also not be required anymore.
3. Figure 2I. Protein level or transcript level? Manuscript says transcript level and figure says protein level. Discrepancy. Stemming from mis-referencing of figures 1 H and 1 I.
4. Fig 1 A, 2 A, 2 F and 3 A DAPI : Fig 1 A says smaller nucleus. Fig 2 A, 2 F and 3 A, do not match this conclusion.
5. Authors could shed more light on what informs their speculation of CB1 receptors being the reason behind contradictory results observed in the present study. (line 424)
The authors have shown the implementation of higher doses of the cannabinoid prior to application of tunicamycin resulted in reduced levels of pro-apoptotic mRNAs, in particular p53, p21, BAX and caspases (Casp3, Casp6 and Casp9), known apoptosis activators.
What was the rationale behind pre-treatment? Why not treatment post ER stress induction? Should similar results be expected in that case?
6. Lines 430 – 435. From the reviewer’s interpretation: Opposing facts presented. How do the authors reconcile these facts. Else, more clarity required.
7. Authors have not discussed the unexpected results upon pre-treatment of 1 and 5 microM THCV and what may be causing this (eg. Fig 3 C, 3 D, 4 J)
Reviewer 2 Report
The research examined how THCV could protect adipose-derived mesenchymal stem cells from endoplasmic reticulum (ER) stress. However, the study solely relied on an in vitro model to assess THCV's effect. The authors should know that in vivo situations are far more intricate than in vitro’s. Hence, relying solely on in vitro data falls short of establishing THCV's protective effect. Therefore, further revisions are necessary as outlined below:
1. Adopt in vivo model to prove the effect of THCV.
2. Although the study's primary focus is on adipogenesis, the authors neglected to investigate whether THCV has any impact on the differentiation of adipose-derived mesenchymal stem cells into adipocytes.
3. Please provide a graphic abstract to summarize the study.
4. Minor problem: in line 287 and line 289, fig 2H and 2I should 1H and 1I.
Round 2
Reviewer 1 Report
Satisfied with the author's rebuttal.
Reviewer 2 Report
Thanks for the revisions made by the authors point by point.